# Scoring Osteoarthritis Reliably in Large Joints and the Spine Using Whole-Body CT: OsteoArthritis Computed Tomography-Score (OACT-Score)

**DOI:** 10.3390/jpm11010005

**Published:** 2020-12-22

**Authors:** Willem Paul Gielis, Harrie Weinans, Frank J. Nap, Frank W. Roemer, Wouter Foppen

**Affiliations:** 1Department of Orthopaedics, University Medical Center Utrecht, Utrecht University, 3584 CX Utrecht, The Netherlands; h.h.weinans@umcutrecht.nl; 2Department of Radiology, University Medical Center Utrecht, Utrecht University, 3584 CX Utrecht, The Netherlands; W.Foppen@umcutrecht.nl; 3Department of Radiology, Central Military Hospital (CMH) Utrecht, 3584 CX Utrecht, The Netherlands; F.J.Nap-3@umcutrecht.nl; 4Department of Radiology, Friedrich-Alexander University Erlangen-Nürnberg & Universitätsklinikum Erlangen, 91054 Erlangen, Germany; Frank.Roemer@uk-erlangen.de; 5Department of Radiology, Boston University School of Medicine, Boston, MA 02118, USA

**Keywords:** computed tomography, image analysis, osteoarthritis, reliability

## Abstract

A standardized method to assess structural osteoarthritis (OA) burden thorough the body lacks from literature. Such a method can be valuable in developing personalized treatments for OA. We developed a reliable scoring system to evaluate OA in large joints and the spine—the OsteoArthritis Computed Tomography (OACT) score, using a convenience sample of 197 whole-body low-dose non-contrast CTs. An atlas, containing example images as reference points for training and scoring, are presented. Each joint was graded between 0–3. The total OA burden was calculated by summing scores of individual joints. Intra- and inter-observer reliability was tested 25 randomly selected scans (*N* = 600 joints). Intra-observer reliability and inter-observer reliability between three observers was assessed using intraclass correlation coefficient (ICC) and square-weighted kappa statistics. The square-weighted kappa for intra-observer reliability for OACT-score at joint-level ranged from 0.79 to 0.95; the ICC for the total OA grade was 0.97 (95%-CI, 0.94 to 0.99). Square-weighted kappa for interobserver reliability ranged from 0.48 to 0.95; the ICC for the total OA grade was 0.95 (95%-CI, 0.90 to 0.98). The OACT score, a new reproducible CT-based grading system reflecting OA burden in large joints and the spine, has a satisfactory reproducibility. The atlas can be used for research purposes, training, educational purposes and systemic grading of OA on CT-scans.

## 1. Introduction

Osteoarthritis (OA) is a leading cause of disability worldwide, with the estimated socioeconomic burden being 1%–2.5% of the gross national product in Western countries [1]. Until now, the search for a disease modifying drug for OA has failed. A key factor for this failure is the use of a one-size-fits-all principle in the development and testing of potential treatments. End-stage osteoarthritis is a fairly uniform disease, but etiological pathways in early disease vary strongly. There is a desire to group OA patients into phenotypes, with the ultimate aim of finding the right treatment for the right patient [2]. The APPROACH study aims to describe these different phenotypes for knee OA and validate models to predict disease progression within these phenotypes [3]. This allows for more patient specific treatments and more efficient clinical trials. The APPROACH study includes knee specific parameters, including patient reported outcome measures (e.g., knee specific questionnaires), physical examination (e.g., knee range of motion), and imaging features (e.g., knee MRI). Additionally, more generic parameters are measured, such as general quality of life, physical performance (e.g., 40 m fast paced walk test) and biochemical marker levels in serum and urine. OA is often a polyarticular disease and the relationship between the latter parameters and knee OA will be heavily influenced by the overall OA burden in the body. However, there is no efficient and standardized method to assess this burden [4,5].

Radiography is widely used for visualizing and grading structural OA. However, it has limited sensitivity for detecting structural damage because of its projectile nature; repeatability is an also issue as positioning errors are common (e.g., wide variations in joint space measurements due to inconsistent flexion of the knee) [6]. Magnetic resonance imaging (MRI) is excellent for visualizing the different tissues within a joint, but it is expensive and time consuming; for example, to obtain good-quality MRI images of multiple joints, the patient would need to lie still for hours. However, CT has several advantages. It uses ionizing radiation to produce a three-dimensional (3D) tomographic images, without the projection limitations of radiography, and is known for its excellent visualization of bone. Advances, such as iterative reconstruction have substantially reduced exposure to ionizing radiation and scanning time [7,8]. Low-dose CT scans provide valuable information on the bony aspects of the joints, with a relatively high signal-to-noise ratio. Whole-body Low-dose CT (WBLDCT) scans, with a scan time of less than one minute and an effective radiation dose <3 mSv for a 70 kg adult male, are increasingly used for evaluation various conditions.

In this study, we aim to develop and describe a WBLDCT-based scoring system to quantify OA burden throughout the body. We believe that the score—the OsteoArthritis Computed Tomography (OACT) score—will be especially useful for research towards personalized OA treatments. We assess the inter- and intra-reader agreement of the new score and present an atlas, with extensive image examples, that can be used for training and educational purposes, for uniform grading of OA on CT-scans.

## 2. Materials and Methods

### 2.1. Study Sample and Image Acquisition

The scoring system was developed using a convenience sample of 197 WBLDCTs acquired for diagnosis or for attenuation correction in PET/CTs in the UMC Utrecht, Utrecht, The Netherlands, between June 2011 and November 2015; the scanning was performed as part of workup for suspected cancer and vascular or infectious disease. Scans were acquired in the supine position without any contrast enhancement, with 64 × 0.625-mm collimation, 120 kV, and dose modulation with a reference of 40 mAs; the estimated effective dose was <3.0 mSv for a 70-kg adult male. Reconstructions in the axial plane were made with 1-mm slices and 0.7-mm increments. Joints with metallic implants were excluded. This study was approved by the local institutional review board (protocol number 15/446-C), with waiver of the need for informed consent.

### 2.2. Image Assessment

The Picture Archiving and Communication System (PACS IDS7 19.3.12; SECTRA) was used to produce multiplanar view reconstructions. Using the 197 scans we created a feasible and reproducible system for grading the severity of OA in each of the major joints. Then, a reference atlas was composed that could be used to teach new readers the scoring definitions. Finally, we tested intra- and inter-observer reproducibility on a subset of 25 randomly selected scans (which included a total of 600 joints).

We aimed to grade all large synovial diarthrodial joints, intervertebral discs (IVD), and facet joints. The elbow was frequently positioned outside the field of view and was therefore excluded. Degenerative disc disease (DDD) of the IVD differs from OA, as IVDs are fibrocartilaginous and not synovial joints. However, the biochemical and radiological features of DDD closely resemble those of OA [4]. Many previous OA studies have assessed the lumbar spine but, as other researchers have suggested, DDD in the cervical and thoracic spine also needs to be considered [9,10]. We first performed a thorough literature search to locate CT-based scoring systems for OA of different joints. If no viable CT-based scoring system was found, we modified the standard radiography–based scores for use on CT images. If no viable scoring system was available for a joint, we developed a new system using the classic radiographic OA characteristics (joint space narrowing, osteophytosis, sclerosis, and subchondral cysts). Each joint was graded on a scale of 0 to 3; thus, four grades were possible. The goal was to develop a scoring system that could be used to score all joints in a single patient within 15 min. The process of development of the scoring system for each joint is described below. The scoring of each joint was discussed in multiple sessions between a group consisting of a MD researcher with 5 years of experiences in medical imaging of OA (WPG), a radiologist in training with a subspecialization in musculoskeletal radiology (WF), and a fellowship-trained musculoskeletal radiologist with 6 years of experience (FJN) and an associate professor, section chief of Musculo-Skeletal Research and attending Radiologist with extensive experience in developing radiologic scores (FWR) The supplementary atlas (Appendix A), which contains extensive examples, can be used for training and also serves as a reference for scoring. Figure 1 presents an overview of the tibiofemoral joint, and Figure 2 shows different grades of tibiofemoral OA.

#### 2.2.1. Upper Extremity

##### Acromioclavicular Joint

Our literature search located a single grading system for acromioclavicular joint degeneration [11]. Using 108 cadaveric joints, Stenlund et al. created a radiographic score that demonstrated satisfactory correlation with macroscopic morphological grade. However, this system was not tested for reproducibility. We used the radiographic characteristics identified by Sterlund et al. to create four grades (Table 1).

##### Glenohumeral Joint

We did not find a validated CT-based grading system for glenohumeral OA. Therefore, we based our score on the widely used and reliable system proposed by Samilson and Prieto that scores OA according to the size of inferior humeral osteophytes on radiographs (Table 1) [12,13]. As CT images offer 3D visualization of the joint, we considered osteophytes everywhere in the glenohumeral joint, i.e., inferior, anterior, and posterior humeral and glenoidal.

#### 2.2.2. Spine

##### Degenerative Disc Disease

The system proposed by Lane et al. for grading degenerative disease of the thoracic and lumbar spine is convenient and reliable [14,15]. We modified it for use on CT images of the cervical, thoracic, and lumbar spine (Table 1). In addition to sclerosis, we considered endplate irregularity, which can be evaluated on CT, as a sign of disease involvement of cartilaginous and bony endplates. Extensive grading 21 spinal levels would be too time consuming, thus, a concise screening of the spine is performed to identify the two most affected levels within the cervical, thoracic, and lumbar regions. For these levels the extensive grading is performed. If these scores are low, this means that degenerative changes in the whole spinal region and therefore we expect limited impact on on systemic biomarker levels and quality of life measurements.

##### Facet Joint OA

We incorporated the grading system created by Weishaupt et al. for the lumbar facet joint OA (an adaption of the original scoring system proposed by Pathria et al.) in our score, extending its application to the cervical and thoracic spine also [16,17].

We recommend the sagittal view for an easier, faster and more reproducible evaluation. Only the two most affected levels within each region are extensively graded (Table 1).

#### 2.2.3. Lower Extremity

##### Hip

Turmezei et al. published a CT grading system for hip OA [18]. This system is highly detailed and time-consuming. In our experience, it takes about 5–10 min for an experienced reader to score 2 hips. The learning curve was long for new readers. We did not find any other grading systems for hip OA on CT and modified the score of Turmezei et al. it to obtain a more straightforward four-grade score based on their principles (Table 1).

##### Knee—Tibiofemoral

We found no validated CT-based grading system for knee OA. A combination of characteristics of radiographic OA as described by Kellgren and Lawrence and, more recently, by Altman et al. (joint space narrowing, osteophytosis, and subchondral cysts) was used to create the four-grade score (Table 1) [19,20].

##### Knee—Patellofemoral

Scoring of patellofemoral joint OA was based on the grades described by Jones et al. [21]. CT is acquired with extended knees, causing the patella to be located proximal to the femoral notch; in this position, it is difficult to accurately measure joint space narrowing. Therefore, we opted for a combined score that considered osteophytosis, sclerosis, and diminishment of the joint space (Table 1).

##### Ankle

The CT scoring system and atlas as published by Cohen et al. was used for grading ankle OA (Table 1) [22].

#### 2.2.4. Total OA Grade

To test the eliability of a total score for OA in the large joints and the spine, a total OA score was calculated by summing the scores of the individual joints. Therefore, with each joint scored on a scale of 0–3, the total score could range from 0 to 72. (Table 1).

### 2.3. Testing Reproducibility

To test intra-observer reproducibility, a medical doctor and researcher with 4 years of experience (WPG) scored the same subset of 25 randomly selected WBLDCTs twice, with an interval of at least 1 week in between. To test inter-observer reproducibility, a radiologist in training, with a subspecialization in musculoskeletal radiology (WF) and a fellowship-trained musculoskeletal radiologist with 6 years of experience (FJN), scored the same random sample of 25 scans independently. The atlas was used as reference for the grading system. In accordance with the Guidelines for Reporting Reliability and Agreement Studies, reliability was tested using Cohen’s kappa for binominal grade, squared weighted kappa for ordinal grade, and two-way intraclass correlation coefficient (ICC) for consistency for the total OA score [23,24]. Kappa values were interpreted according to Landis and Koch: i.e., 0–0.20 slight agreement; 0.21–0.40 fair agreement; 0.41–0.60 moderate agreement; 0.61–0.80 substantial agreement; 0.81–1 almost perfect agreement [25]. Agreement was tested using absolute agreement percentages for binominal and ordinal grades and Bland–Altman and Jones plots for continuous values [26,27]. All analyses were carried out in R version 3.4.4 (https://cran.r-project.org/) using the irr package, version 0.84.

## 3. Results

The 197 scans used for the development of the atlas were acquired from a sample comprising 43% males (85/197). The mean age (SD) of the patients was 54 (±15) years. Indications for scanning included vasculitis (*n* = 106), suspected infection (*n* = 57), and suspected malignancy (*n* = 34). The 25 scans included in the reliability analyses were from a patient subset that comprised 44% males (11/25). The mean age (SD) of the patients was 54 (±17) years. Indications for scanning were vasculitis (*n* = 15), suspected infection (*n* = 8), and suspected malignancy (*n* = 2). Within the test set, OA grades 0 to 3 were found in all joints, except for the hip and ankle, where only grades 0 to 2 were found (Table 2). Most joints were graded as having no OA or only mild OA, which is to be expected in a random sample of hospital. One ankle could not be scored due to beam-hardening artifacts caused by screws.

### 3.1. Intra- and Interobserver Reliability for Total OA Grade

Intra-observer reliability for total OA grade was excellent, with an ICC of 0.97 (95% CI, 0.93 to 0.99). The Bland–Altman plot showed an even spread of errors between the first and second observation, with a mean error of −3.5 (SD, 3.4). Inter-observer reliability for total OA grade was also excellent, with an ICC of 0.94 (95% CI, 0.86 to 0.98). ICCs for inter-observer reliability were comparable between observer pairs of different proficiency levels, 0.95 between WPG and WF, 0.93 between WPG and FJN, and 0.97 between WF and FJN. The Jones plot showed an even spread of errors between all observers, with WF giving grades around the mean, FJN giving lower grades on average, and WPG giving higher grades on average (Figure 3).

### 3.2. Intra- and Interobserver Reliability for OACT Scores for Individual Joints

Intra-observer reliability of the OA grades for individual joints was substantial to almost perfect, with the kappa values ranging from 0.79 to 0.95 and absolute percentage agreement, ranging from 67% to 92% (Table 3). Inter-observer reliability of the OA grades for individual joints was moderate to almost perfect, with the kappa values ranging from 0.48 to 0.95 and absolute percentage agreement ranging from 36% to 90% (Table 3). Table A1 shows the intra- and inter-observer reliability for grading of individual OA characteristics (joint space narrowing, osteophytosis, and so on).

## 4. Discussion

The OACT score described here—a new reproducible WBLDCT-based grading system for OA in large joints and the spine—was developed for research purposes. In this first step, we introduce the scoring methods and present a reference atlas with multiple example images. The atlas can be used as a reference for training new readers, educational purposes and systemic grading of OA on CT-scans. We demonstrated a satisfactory intra-observer reliability and decent inter-observer reliability. The use of WBLDCT for this goal is associated with short scanning time with comparatively low-level exposure to ionizing radiation (effective radiation dose <3 mSv for a 70-kg adult male). Furthermore, with this newly developed grading system, it is possible to reliably assess overall structural burden of OA in a patient within 15 min.

There is still no disease modifying drug for OA, mainly because drug development focused on finding a one-size-fits-all drug. Drug development and evaluation will have a higher chance of success if it is focused on specific structural phenotypes of OA. The selection criteria for these OA phenotypes has to be determined. The APPROACH study uses a combination of established and novel biomarkers to develop stratification models that can help select the appropriate therapy for each knee OA patient [3]. Many parameters, such as quality of life, physical performance and biochemical markers levels in serum or urine are affected by the disease burden of other joints [4,28,29,30,31]. These parameters potentially impact the efficacy of drug development and evaluation in OA. In the APPROACH study, the OACT score helps to phenotype OA patients and correct for confounding at the patient level when assessing the relation between systemic biomarkers, and e.g., knee OA. Besides structural progression, disease burden is an important marker for treatment success. Eventually the OACT-score will help improve patient selection for OA observational studies and clinical trials that include clinical outcome parameters. The clinical relevance needs to be established before clinical application may be considered. This has been the case for many other scoring-based assessment instrument in the field of OA that were primarily developed in the context of MRI evaluation [32,33]. Future studies should test the validity of the OACT-score against clinical outcome parameters and other biomarkers.

In our sample the total OACT score showed excellent intra- and inter-observer reliability (ICC, 0.97, and 0.94, respectively). To our knowledge, this is the first study test to reliability for an OA grade at patient level. However, we would like to stress that summing separate ordinal grades has limitations; for example, this would result in multiple low-grade joints being equivalent to a single high-grade joint. For future studies, the weighting factors for composing a total score, reflecting OA throughout the body, should be altered to the goal of the specific study. Systemic cartilage degradation markers or global quality of life measurements could be used to assess the influences of the different joints on the total OA burden in future studies. Adding the OA scores of the joints of the hands and feet would undoubtedly improve the value of the scoring system; however, we did not do so because of the variable positioning of the hands and feet in the CT images in our study. Validated radiographic scores for OA of the hands and feet could be used in combination with the OACT score for a more complete assessment of total OA burden in the body [34].

The reliability results are in the expected range for a semi-quantitative radiological score for OA. For the acromioclavicular joint, we found substantial to almost perfect reliability. No other CT-based study is available for comparison. For the glenohumeral joint, inter-observer reliability was moderate to substantial, while the intra-observer reliability was almost perfect. We expect the moderate intra-observer reliability to be caused by the high prevalence of no and mild glenohumeral OA, as this emphasizes the decision between the presence of no, or a small (<3 mm) osteophyte. Again, no CT-based studies are available for comparison. We found almost perfect intra-observer reliability and substantial to almost perfect inter-observer reliability for DDD. No CT-based studies are available for comparison. While, OA and DDD are different entities, the response to mechanical loading, symptoms and matrix degradation pattern are highly correlated [35]. Therefore, we chose to include DDD in our score. Based on the aim of their study, researcher may decide to in- or exclude DDD.

Pathria et al. tested the inter-observer reliability of their CT-based scoring system for facet joint OA and reported a kappa value of 0.46, while Weishaupt et al. reported a weighted kappa of 0.60 [16,17]; the overall percentage agreement was 63%, and 51%, respectively. These results were comparable to our results, where the weighted kappa values ranged from 0.66 to 0.68 and absolute percentage agreement ranged from 57% to 64%.

Turmezei et al. tested the reliability of their CT grading system for hip OA and reported a weighted kappa of 0.74 and 0.75 for intra- and inter-observer reliability, respectively. We simplified their scoring system to enhance grading speed and reliability for new readers and found a weighted kappa of 0.85 for intra-observer reliability and between 0.48 and 0.65 for inter-observer reliability. The lower inter-observer reliability in our study may be due to the very low prevalence of hip OA in our study population (8% with moderate OA or higher) compared to the study population of Turmezei et al., which was selected to include the full spectrum of hip OA.

For both patella and knee OA, we found almost perfect intra-observer reliability and substantial to almost perfect inter-observer reliability. For the ankle joint, we found moderate to substantial inter-observer agreement. Cohen et al. introduced an atlas for grading ankle osteoarthritis on CT and reported an ICC of 0.851 and unweighted kappa of 0.582 in a population of specifically selected scans. As such, a valid comparison with our results is not possible.

Our scoring system has several limitations. First, it does not consider OA in the elbows, hands, and feet. The elbow was not included in our score as it was positioned outside the field of view in a large number of scans. However, it should be noted that elbow OA is rare, with a prevalence of only ~2% [36]. Second, we used semi-quantitative grades. However, it must be noted that semi-quantitative grading enabled scoring a full WBLDCT in 15 min. Third, WBLDCT is obtained with the patient lying supine; assessment of joint space is influenced by the lack of weight bearing. The development of weight-bearing CT-scan will hopefully counter this problem in the near future. Fourth, WBLDCT can clearly visualize bony changes, but soft tissue degeneration (e.g., meniscal and capsule tears) will be missed. Fifth, concurrent pathology such as diffuse idiopathic skeletal hyperostosis may aggravate OA scores. Grading systems for such concurrent diseases could be used along with the OA scores to further characterize individuals [37,38,39]. Sixth, CT involves exposure to possibly harmful ionizing radiation. Due to technical advances, including iterative reconstruction, the effective radiation dose of the WBLDCT was around ≤3 mSv, which approximates one year of background radiation [40]. The exact risk for excess death by cancer to a given effective radiation dose is difficult to determine. Using the rule of 5% excess mortality per 1 Sv, each WBLDCT may be accompanied by a 0.00015% excess risk for cancer mortality [41]. Determining the sample size for a reproducibility study using weighted kappa statistics is not straightforward [24]. We deemed a sample of 25 as appropriate since this results in a minimum of 50 joints per analysis and a total time invested for training and scoring of ~10 h per reader. For the analysis of the total OA grade, only 25 cases were available, which partly explains the high standard deviations in the Bland–Altman and Jones plots.

## 5. Conclusions

To summarize, we introduce the OACT score, a WBLDCT-based reproducible grading system for large-joint OA burden in the body. The OACT score can be used as an outcome measure in OA research or to correct for the influence of total OA burden on patient reported outcomes and biochemical marker levels.

## Figures and Tables

**Figure 1 jpm-11-00005-f001:**
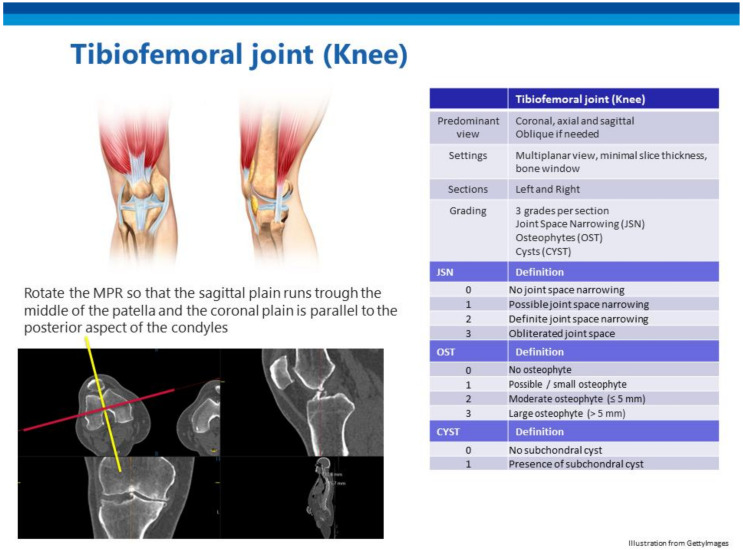
An example from the atlas showing the overview for scoring tibiofemoral osteoarthritis.

**Figure 2 jpm-11-00005-f002:**
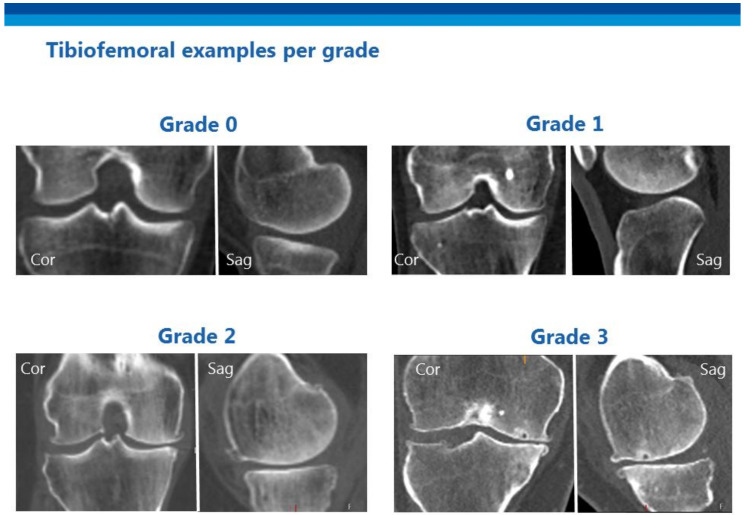
Example images from the atlas showing different grades of tibiofemoral osteoarthritis.

**Figure 3 jpm-11-00005-f003:**
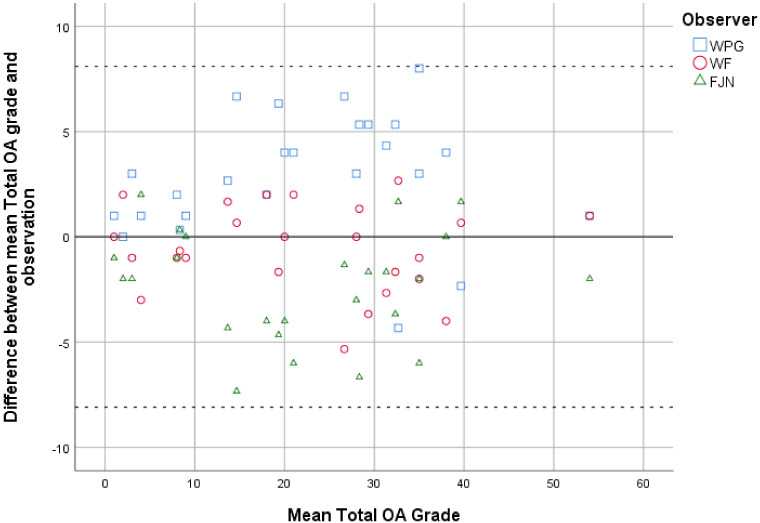
Jones plot depicting the difference between each observation of the different readers and the mean observed score for the total OA grade. The interrupted lines show the 95% limits of agreement.

**Table 1 jpm-11-00005-t001:** Definition of OACT scores for individual joints.

**Acromioclavicular joint**	
0	No osteophytes or joint space narrowing (JSN)
1	Lipping and/or possible JSN
2	Definite osteophytes and/or JSN
3	Definite osteophytes and/or JSN and sclerosis and/or cysts and/or bony deformities
**Glenohumeral joint**	
0	No osteophytes or definite JSN
1	Osteophyte measured less than 3 mm
2	Osteophyte measured between 3 and 7 mm, slight joint irregularity
3	Osteophyte measured more than 7 mm, definite JSN and/or irregularity.
**Degenerative disc disease**	
0	Score 0–2 (Based on disc space narrowing, osteophytes, end plate regularity and sclerosis)
1	Score 3–5
2	Score 6–8
3	Score 9–10
**Facet joint**	
0	Normal facet joint space width (JSW) (2–4 mm)
1	Narrowing of facet JSW (<2 mm) and small osteophytes and/or mild hypertrophy of the articular process
2	Narrowing of facet JSW (<2 mm) and moderate osteophytes and/or moderate hypertrophy of the articular process and/or mild subarticulare bone erosions
3	Narrowing of facet JSW (<2 mm) and large osteophytes and/or severe hypertrophy of the articular process and/or severe subarticulare bone erosions and/or subchondral cysts
**Hip joint**	
0	Score 0–1(Based on joint space narrowing, osteophytes, and cysts)
1	Score 2–3
2	Score 4–5
3	Score 6–7
**Tibiofemoral joint**	
0	Score 0–1(Based on joint space narrowing, osteophytes, and cysts)
1	Score 2–3
2	Score 4–5
3	Score 6–7
**Patellofemoral joint**	
0	No osteophytes, joint space narrowing (JSN)/sclerosis
1	Small osteophyte/lipping and mild JSN, but no defined sclerosis
2	Moderate osteophytes, moderate JSN and possible sclerosis
3	Large osteophytes, (near) boney contact and defined sclerosis
**Ankle joint**	
0	No clinical evidence of OA; joint space integrity fully intact
1	Mild; osteophyte formation/lipping, possible joint space narrowing
2	Moderate; joint space narrowing evident, obvious osteophyte formation and some sclerosis/cystic changes
3	Severe; near absence of joint space, severe osteophyte/cyst formation, deformity of bone

All subscores are presented in the atlas.

**Table 2 jpm-11-00005-t002:** Frequency of grades per joint (*n* = 25 patients).

Joint	0 (No)		1 (Mild)		2 (Moderate)		3 (Severe)	
Acromioclavicular, N(%)	24	(48)	10	(20)	5	(10)	11	(22)
Glenohumeral, N(%)	37	(74)	7	(14)	3	(6)	3	(6)
Intervertebral Disc, N(%)	48	(32)	47	(31)	33	(22)	22	(15)
Facet, N(%)	91	(61)	37	(25)	7	(5)	15	(10)
Hip, N(%)	33	(66)	13	(26)	4	(8)	0	(0)
Knee, N(%)	25	(50)	13	(26)	8	(16)	4	(8)
Patellofemoral, N(%)	25	(50)	15	(30)	5	(10)	5	(10)
Ankle^1^, N(%)	26	(54)	19	(38)	4	(8)	0	(0)

^1^ One ankle was not scored due to artefacts caused by screws; Scores presented are produced in the first scoring round by WPG.

**Table 3 jpm-11-00005-t003:** Intra- and interobserver reliability as weighted kappa (percentage of absolute agreement) for OACT scores for individual joints.

Joints	Reader 1 (intra)		Reader 1 vs. Reader 2		Reader 1 vs. Reader 3		Reader 2 vs. Reader 3	
Acromioclavicular	0.84	(80)	0.87	(74)	0.75	(62)	0.82	(68)
Glenohumeral	0.95	(92)	0.69	(72)	0.58	(38)	0.50	(48)
Intervertebral Disc	0.85	(67)	0.80	(61)	0.80	(68)	0.77	(53)
Facet	0.90	(85)	0.68	(64)	0.66	(57)	0.66	(57)
Hip	0.85	(88)	0.53	(68)	0.65	(64)	0.48	(64)
Knee	0.84	(72)	0.85	(68)	0.73	(50)	0.64	(36)
Patellofemoral	0.94	(88)	0.95	(90)	0.79	(60)	0.78	(64)
Ankle	0.79	(84)	0.74	(80)	0.56	(65)	0.49	(63)

Reader 1: Medical doctor and researcher; Reader 2: Radiologist in training with a subspecialization in musculoskeletal radiology; Reader 3: Fellowship-trained musculoskeletal radiologist with five years of experience.

## Data Availability

The data presented in this study are available on request from the corresponding author. The data are not publicly available due to ongoing unpublished research.

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
