# Peer review of "Scoring Osteoarthritis Reliably in Large Joints and the Spine Using Whole-Body CT: OsteoArthritis Computed Tomography-Score (OACT-Score)"

_jpm, 2020, doi:10.3390/jpm11010005_

Round 1
Reviewer 1 Report
Well done article with an excellent explanatory and high quality atlas. there was no such accurate score for assessing the degree of osteoarthritis of the large joints and spine
Reviewer 2 Report
This article introduced a newly developed method to evaluate morphological progression of large joint including spine for patient with osteoarthritis using whole-body CT. The scoring system was elegant and the results were convincing for both of reliability and reproducibility.
One thing I want the author to add in discussion is a problem of radiation exposure. It might be quite large exposure for patient when this scoring system is popularized. The author should describe regarding this problem.
Reviewer 3 Report
Review of the manuscript entitled: “Scoring osteoarthritis reliably in large joints and the spine using whole-body CT: OsteoArthritis 3 Computed Tomography-score (OACT-score)”
In this manuscript the authors introduce a standardized method to assess structural osteoarthritis (OA). They aimed to develop a scoring system to evaluate OA in large joints and in the spine. To that end, they used the convenience sample of 197 whole-body low-dose non-contrast CTs. They assessed the intra- and interobserver reliability in 25 randomly selected scans. They found that square-weighted kappa for the intraobserver reliability ranged from 0.79 to 0.95 and the intraclass coefficient (ICC) for the total OA grade was 0.97. Square-weighted kappa for the interrater reliability ranged between 0.48 and 0.95 and the ICC was 0.95. they concluded, that the OACT score as a satisfactory repeals reproducibility.
This is a carefully executed study, dealing with an important subject. As the authors state, to date there is no standardized method to assess structural osteoarthritis. Thus, it is an commendable endeavour to implement a scoring system for OA for research and educational purposes.
I only have a few questions/comments:
Abstract:
In the abstract, it should be stated, how many raters were involved.
Methods:
Please consider indicating the institution where the CT scans took place.
L87 “Finally, we tested intra- and interobserver reproducibility on a subset of 25 scans (which included a total of 600 joints).”
How were the 25 scans selected from the initial sample of 197 patients? Where they chosen randomly?
Do you think it is sufficient, the tests the intraobserver reliability with only one observer?
I noted some similar examples in the literature, yet I am not yet convinced that one can draw definite conclusions on a test based on the observation done by a single person.
Discussion:
The superordinate aim of research on structural OA is to find ways to relieve pain and to improve function. How does the OA score correlate to these aims? Please consider inserting a paragraph discussing this item.
L235 “Many parameters, such as quality of life, physical performance and biochemical markers levels in serum or urine are affected by the disease burden of other joints.” Can you give references for this statement?
L239 “Major advantages of using WBLDCT for this goal, are the short scanning time and the low-level exposure to ionizing radiation (effective radiation dose <3 mSv for a 70-kg adult male).”
Consider inserting “comparatively” after “low-level”.
I would not consider exposure to ionizing radiation as an advantage. Moreover, as pointed out in the introduction, higher sensitivity for detecting structural damage should be mentioned here.
Radiation exposure: I agree, that the required radiation exposure is tolerable. However, this should be discussed more in detail.
